# Adolescent choices and caregiver roles: Understanding individual and interpersonal influences on sexual decision-making in South Africa

**Heeran Makkan** [1,2¤]*, **Yvonne Wangui Machira**[3], **Funeka Mthembu** [1], **Omphile Masibi**[1], **Thuso Molefe** [1], **Pholo Maenetje** [1,4], **Vincent Muturi-Kioi** [3], **Matt A. Price** [3,5], **Vinodh Aroon Edward** [1,2,4,6], **Candice Chetty-Makkan** [1,7,8]

1 The Aurum Institute, Rustenburg Research Centre, Rustenburg, South Africa, 2 Department of Interdisciplinary Social Science, Public Health, Utrecht University, Utrecht, The Netherlands, 3 IAVI Africa, Nairobi, Kenya, 4 Department of Medicine, Vanderbilt University, Nashville, Tennessee, United States of America, 5 Department of Epidemiology and Biostatistics, University of California at San Francisco, San Francisco, California, United States of America, 6 School of Health Sciences, College of Health Sciences, Westville Campus, University of KwaZulu-Natal, Durban, South Africa, 7 Health Economics and Epidemiology Research Office (HE2RO), Wits Health Consortium, Johannesburg, South Africa, 8 Faculty of Health Sciences, University of Witwatersrand, Johannesburg, South Africa

¤ Current address: The Aurum Institute, Johannesburg, South Africa
* hmakkan@auruminstitute.org

## Abstract

South African adolescents are at-risk for HIV infection due to engaging in high-risk sexual behaviours. Understanding the factors influencing sexual decision-making is crucial for developing effective HIV prevention strategies. We conducted a qualitative study with adolescents and caregivers in Rustenburg, South Africa to explore individual and interpersonal factors that influence adolescent sexual decision-making. Focus Group Discussions (FGDs) were conducted in English and Setswana with 17 adolescents (13 females and 4 males) and 19 caregivers (17 females and 2 males) between April and July 2018. Thematic analysis revealed that while adolescents had access to sexual education from various sources, this knowledge did not translate into healthy sexual decision-making. Lack of effective communication and support between caregivers and adolescents in discussing sexual behaviours are a contributing barrier. Although adolescents expressed a strong need to be understood and supported by caregivers regarding sexual behaviours, there was perceived distrust, judgemental attitude from caregivers, poor role models of a father figure, and the traditional taboo nature of having these discussions with caregivers. While female adolescents and female caregivers discussed sexual matters, this type of communication was limited with adolescent males. Male adolescents were uncomfortable communicating with either caregiver, fearing caregiver judgemental attitudes and being misunderstood. Female caregivers perceived male caregiver roles to be absent and non-engaging. Caregivers desired to support their children, yet they seemed to doubt their skills. Communication tools and guidance on how adolescents and caregivers could communicate about sexual matters could create enabling environments for adolescents to make informed, healthy decisions

**Data Availability Statement:** Data can be found on a Figshare link, https://doi.org/10.25382/iavi.26540635.v1.

**Funding:** This study is part of the ADVANCE program. The ADVANCE program, led by IAVI, is a cooperative agreement (#AID-OAA-A-16-00032) made possible by the support of the American People through the U.S. President's Emergency Plan for AIDS Relief (PEPFAR) through the United States Agency for International Development (USAID). The funders had no role in study design, data collection and analysis, decision to publish, or preparation of the manuscript.

**Competing interests:** The authors have declared that no competing interests exist.

regarding their sexual behaviours. Further, future interventions could consider gain-framed messaging to address adolescents translating knowledge of sexual behaviours to making healthy sexual choices. Improving equitable male caregiver role at home, is of particular importance in supporting adolescent sexual decision-making, and should be prioritized.

## Introduction

The HIV burden among Sub-Saharan African (SSA) adolescents is a significant public health concern with adolescent girls and young women (AGYW) aged 15–24 years accounting for 63% of new HIV infections in the region [1]. In South Africa, AGYW are particularly vulnerable having an HIV acquisition rate three times higher than their male counterparts [2]. This vulnerability is exacerbated by various factors including sexual relationships with older men, gender inequalities, social norm discriminatory practice, poverty, deprived autonomy, and risk for abuse and violence [3–5]. Although adolescent boys and young men (ABYM) are at lower risk for HIV acquisition, there is a growing need to include ABYM in HIV prevention discussions [1, 4]. Therefore, it remains vital to understand both AGYW and ABYM perspectives regarding their sexual decision-making and risk for HIV [1, 4, 6].

In South Africa, several studies have been conducted with adolescents to understand participant preferences for HIV prevention modalities [7–10]. Many studies have focused on identifying barriers, motivators, and preferences to improve the uptake and persistent use of HIV prevention modalities, which are usually designed without including the voices of adolescents [7, 8]. This important step of including adolescents from the onset during the co-design phase of HIV prevention interventions is essential, yet often overlooked.

Importantly, in South Africa and elsewhere in SSA, engagement with caregivers (parents or guardians) is also rare when it concerns adolescent sexual decision-making [11–13]. Although caregivers are uniquely positioned to educate and counsel their adolescents on how to make healthy sexual decisions only 20% of South African adolescents receive sexual health education from their caregivers [13, 14]. The role of discussing sexual matters with adolescents generally becomes the responsibility of the female caregivers, as male caregivers are generally perceived as being absent, difficult, uncomfortable, and incapable of fostering close bonds with children [13, 14]. Although there are benefits of improving the adolescent-caregiver communication to promote informed decision-making in adolescent sexual behaviours, cultural barriers, long work hours, and limited knowledge and experience in reaching their children further impede the communication between adolescents and their caregivers [11–14].

The socioecological model (SEM) provides a framework for understanding the multiple levels of interactions that could influence adolescent sexual decision-making [15–18]. Applying the SEM to explore the perspectives of adolescents and others in their social ecology can provide valuable insights into the complex interplay of factors shaping adolescent sexual decision-making [16, 18, 19]. While there are several studies on adolescent sexual behaviour and behavioural interventions, there is a dearth of information on understanding the underlying individual and interpersonal factors which influence healthy decision-making among adolescents [16, 18, 19]. Although several intervention strategies were identified from past studies, limited studies explore how the role of caregivers could influence adolescent's ability to make healthy sexual decisions [19]. Therefore, we conducted a qualitative exploratory study to understand the factors that influence sexual decision-making from the perspectives of adolescents and their caregivers in Rustenburg, South Africa.

## Methods

### Ethics statement

The study was approved by the University of the Witwatersrand, Human Research Ethics Committee (ref. no 170607), and the Research Committee of the Northwest Provincial Department of Health. All participants were given a small token of appreciation ZAR150.00 (about $8.20 USD) for sparing their time to take part in the study and to compensate for any expenses they may have had.

Adolescents who were between 18–19 years provided written informed consent while we received assent from those who were younger than 18 years and obtained written caregiver consent. Caregivers consented to take part in the FGDs. All participants provided consent for the FGDs to be digitally recorded. Illiterate participants provided a thumbprint to acknowledge understanding in the presence of a witness. Participants consented to their direct quotations being published.

### Study design and settings

This qualitative project was embedded in a parent observational prospective study that took place at the Aurum Rustenburg Research Centre, South Africa. Rustenburg is one of the fastest growing cities in the Northwest Province where platinum mining is the dominant industry, with an approximate population of 562,031 in 2022, where 25% being 15 years old or younger [20].

The parent study has been previously reported [21], and was conducted from April 2018 to November 2020, where we recruited males and females aged between 12 to 19 years, who were HIV-negative and in females not pregnant. The parent study aimed to evaluate adolescent preparedness for HIV prevention trials, assess clinical trial knowledge, measure willingness to participate in HIV prevention research, and characterize adolescent HIV risk and sexual decision-making. Adolescents who met the eligibility criteria for the parent study and the caregivers (parent or guardian) of 15- to 17-year-old eligible participants were invited to take part in focus group discussions (FGD). FGDs were conducted within three months after enrolling into the main study, between April to July 2018. Adolescents and caregivers were selected using convenience sampling on a first come basis, considering their availability for a scheduled date within the 3-month window, and concluded once a quota of 8 to 10 participants were reached. Caregivers were not necessarily the caregivers of adolescent participants who participated in FGDs.

### Data collection

Two in-person FGDs were conducted with adolescents and two with caregivers following enrolment into the main study and before a month-3 follow-up visit. Each FGD comprised 8–10 participants. The adolescent FGDs were disaggregated by age only (15–17 years and 18–19 years). FGDs were conducted in English and Setswana (the predominant local language in Rustenburg) and took approximately one hour.

FGD guides (S1 and S2 Text) were used to explore the adolescent and caregiver knowledge of sexual and reproductive health, and factors contributing to sexual and reproductive health behaviour. The guides included many topics, including to understand the factors that influence sexual decision-making from the perspectives of adolescents and their caregivers and recommendations which could encourage adolescents to speak honestly about their sexual behaviours. We identified deductively from literature that could potentially inform sexual decision-making. Probes under these broad categories were used to understand adolescent

sexual decision-making. To measure saturation, we reviewed probes in real time and adjust probing to ensure that we gathered adequate information across all participants.

Research assistants OM and TM (graduate education), female and male respectively, were the facilitators and note takers. They were trained by an experienced post-doctoral investigator (CCM) before data collection. During each FGD, research assistants OM and TM (graduate education) served as either facilitators or note takers and positioned themselves across from one another within the discussion circle. Facilitators and note takers (OM, TM, FM and CCM) had professional relationships with participants and their caregivers, which ended post the main study conclusion.

During screening into the parent study, adolescents completed an interviewer-administered socio-demographic and sexual behaviour questionnaire. Caregivers completed an abbreviated interviewer administered socio-demographic questionnaire only. We used data from the socio-demographic questionnaires to describe our participant pool for this study.

### Data analysis

All FGDs were digitally recorded, transcribed verbatim and translated to English (where necessary) prior to analysis by OM and TM. Following each FGD and after the debrief sessions, the digital recordings were reviewed to revise the probes and monitor the saturation of themes. FM conducted a quality check of the transcripts. De-identified English transcripts and debrief notes were also reviewed before data analysis and included in the codebook. NVIVO software Version 14 was used for the analysis [22]. For the analysis, HM used a thematic approach that included deductive and inductive approaches to derive the themes. Through inter-rater reliability (YWM and CCM), we reached an agreement on the coding and emerging patterns. Discrepant codes were dropped during the analysis. Emerging themes were further discussed and finalised during virtual meeting discussions with the research team. We applied a 32-item checklist (S1 Checklist), the Consolidated Criteria for Reporting Qualitative Research (COREQ), to ensure comprehensive reporting of qualitative data [23].

## Results

### Descriptive characteristics of participants

In total, 17 adolescents (13 females and 4 males) participated in the two FGDs, with median age 17-years (Interquartile range (IQR) 16–18 years). Most adolescents (n = 13, 76.5%) attended senior secondary school. Nine females and one male reported being in a relationship with a steady partner/s, but only the females reported being sexually active. Five (55.6%) of these nine females reported multiple heterosexual sexual relationships with older partners up to four years older than them. Sixteen (94.1%) adolescents reported alcohol use, at a minimum once per month.

Seventeen female and 2 male caregivers with a median age of 37 years (IQR 34–46 years) took part in the two caregiver FGDs. Thirteen were unemployed, with twelve reliant on family or spouse/partner financial assistance. All caregivers had about 2 to 5 child dependents per household. Among the 17 female caregivers, seven were married, and seven were in a steady non-cohabiting relationship. Thirteen female caregivers had their first child before they were 21 years old (range 16–20 years old).

### Qualitative findings

Table 1 lists the themes aligned with the individual and interpersonal levels from the SEM.

**Table 1. Emerging themes that influence adolescent sexual decision-making from adolescent and caregiver perspectives in Rustenburg, South Africa.**

| Social Ecology Level | Themes | Description |
|---|---|---|
| Individual level factors influencing sexual decision-making | Gaps in theoretical knowledge and actual behaviour | Theoretical knowledge source is from school curriculum (educators), friends, digital media, social media, television and radio; Adolescents know of sexual risk-behaviours and methods to prevent negative outcomes yet continue to engage in sexual behaviours which may have negative health outcomes. |
| | Age milestone informed ability to become sexually active | Reaching adulthood, puberty, body development and traditional beliefs are factors contributing to the perception when an adolescent may become sexually active |
| | Alcohol consumption by adolescents and their exposure for high-risk sexual activities | Adolescents often consume alcohol when engaging in "fun" activities and lose their inhibition—resulting possible exposure to high-risk sexual behaviour, and sexual and violent outcomes |
| Relationship level factors influencing sexual decision-making | Caregiver roles in educating and counselling adolescents about sexual behaviour | Caregivers and adolescents have comfort in discussing sex topics with a person of the same gender. Female caregivers are often left to play this supportive role either in single mother or duo caregiver homes. |
| | Importance of the male caregiver figure | Male caregivers are needed to have an active role in the upbringing of their children, especially to when discussing sex topics with a male adolescent. Male caregivers in contrast felt they could not perform this function as their sons were much closer to their mothers. |
| | Caregiver and adolescent perceptions of communication barriers | Both adolescents and caregivers indicated barriers in communication to be judgemental attitudes, distrust, lack of experience and skill, and traditional taboo nature in engaging in sex discussions with children or caregivers. |

**Individual level factors influencing sexual decision-making.** Adolescents received information on sexual behaviour and how to make sexual decisions from school, friends, partners, television and social media that were confirmed in responses from the caregivers. Their exposure to social media created a sense of peer pressure in a need to share naked pictures of themselves, as did many of their peers.

*". . .sexting whereby you text your partner, talking about those things. . .I learned mostly about these things at school and from friends when they tell you that they saw this and that on TV and also from cell phones* [smart phone device] *and Facebook."*

Female, 16 years old, not sexually active.

*"Social media influences us because on Facebook you see other children who are younger than you are posting naked pictures of themselves then you ask yourself why are you not posting naked pictures of yourself when you older than them. . .So that influences you to grow up."*

Male, 15 years old, not sexually active.

*"On the TV a lot of the times we see. . .they watch sexual movies on the TV right. So that's the thing that encourages them to end up being a part of that thing".*

Female caregiver, 34 years old, single with steady partner.

However, adolescents and caregivers perceived that there is a gap in translating the knowledge of sexual behaviour into healthy sexual decisions. This an action meant to prevent the sexual behaviour which may result in HIV infection or pregnancy.

*"They know about risky behaviour. . . It is just that they are hard-headed".*

Female, 18 years old, sexually active, single with multiple partners.

"*I don't think that all adolescents know because if they knew they wouldn't engage in unprotected sex. . .become pregnant when they are young, they are getting HIV. . .*".

Male caregiver, 46 years old, married.

Reaching a certain age milestone informed adolescent decision-making to becoming sexually active. Adolescents had polarized views though, where some preferred younger age associating this with the onset of puberty while others preferred waiting until adulthood.

"*15 years because a lot of things would have changed on your body, like the eggs in a women's' body.*",

Male, 15 years old, not sexually active, no partner.

"*I think at 12 because at 12 you starting to be matured. For puberty starting.*"

Female, 19 years old, sexually active, and has a single partner.

"*. . .a person should start having sex at the age of 21, because 21 you are mature enough, you are able to take right decisions when coming to sex.*"

Female, 19 years old, sexually active, single partner.

Alcohol use was perceived to promote engagement in high-risk sexual behaviour, voluntarily and involuntarily, reducing the ability to make informed healthy decisions.

"*Risky behaviour according to my understanding, is like when we go out with friends and have fun and start behaving in a way that we start drinking and not seeing the outcomes. . ..*"

Female, 18 years old, sexually active, multiple partner relationship.

"*. . .some girls. . .while they are sitting there at the taverns with the guys, the girl would go to the toilet and the other guys will put a drug in the girls drink. Then the girl drinks it coz* (because) *she has no idea of what they put inside of her drink. Whatever they put inside of your drink, drugs you and you become easily influenced into doing anything that they want you to do.*"

Male, 15 years old, not sexually active.

Poverty and the need for financial support were highlighted as other factors influencing the decision to engage in sexual behaviours.

"*Sometimes what leads some adolescents to sexual activities is that you get that at home they are in need of something so the girls get forced to turn to older men for financial support. . .-which leads them to sexual activities.*"

Female, 18 years old, sexually active, single with multiple partners.

"*They take them to be "stupid" . . . than they end up falling in that trap, and again. . . Sometimes you get {that} the young girls hang out with us older men, and we give them money.*"

Male caregiver, 50 years old, married.

**Relationship level factors influencing sexual decision-making.**  Female adolescents believed a female caregiver, being the same gender would identify with them. In contrast,

some female adolescents shared their greater comfort in communicating with their male caregiver. Others expressed comfort in talking with siblings.

> "*I prefer to talk to my mother... [she] is a woman, and she will understand.*"

Female, 19 years old, sexually active, in a single partner relationship.

> "*I honestly speaking I trust my father and I can talk to him about anything...We have a very close relationship. My mother can't keep a secret, but my father can.*"

Female, 18 years old, sexually active, in a single partner relationship.

> "*Sister. I cannot to talk to my mother or my father.*"

Female, 19 years old, sexually active, in a single partner relationship.

Female caregivers described how the role of a male caregiver or a father figure can create barriers to communication between adolescents and caregivers. Female caregivers believed that male caregivers ignored their supportive role in favour of fulfilling personal desires. In contrast, male caregivers may feel it is futile to reach out to sons because sons tend to develop closer bonds with female caregivers.

> "*...is just that in most cases men are ignorant... he comes home tired, reads newspaper, watch news and sleeps... or is at the gym. I am the one that is always with the children*".

Female caregiver, 45 years old, married.

> "*With fathers it is not that easy... even if they are boys, they are close to their mothers* [agreement by other caregivers]... *they are even scared to talk to us...*".

Male caregiver, 46 years old, married.

While caregivers expressed their desire to support their children, many adolescents were resistant to this type of engagement due to concerns citing trust, confidentiality, and feelings of being misunderstood. Male adolescents expressed difficulty in speaking with both male and female caregivers, fearing caregiver judgemental attitudes and being misunderstood.

> "*We must be open... We have to be free but not too much right? We have to try to talk to them... our parents did not talk to us... So nowadays, we have to... teach the right things*".

Female caregiver, 49 years old, married.

> "*I do not talk with them, because they are going to think that you have already started having sex at a young age.*"

Male, 16 years old, not sexually active.

> "*I do not talk to them because like they won't understand what I'm going through*".

Male, 15 years old, not sexually active.

However, some caregivers and adolescents avoided discussions on sexual behaviour due to the historically taboo nature of talking about sex or other sexuality topics with minors.

*"Our parents were hiding things from us, we . . . just lived in the dark. Today if you tell a child something, they will say "yes in your times". So, we should find a balance and not put too much pressure. . . We must try to keep up with their times."*

Female caregiver, 49 years old, married.

*". . .according to my culture at my age I cannot talk to my parents about sex.. . .I prefer to talk to my friend because she broke her virginity before me, and she never told me to do the same. I decided to break my virginity. When she gives me advice, I will listen and see for myself if it is right or wrong."*

Female, 19 years old, sexually active with a single partner.

Adolescents and caregivers expressed interest in improving communication on sexual behaviour and engaging in sexual behaviour discussions, to improve healthy sexual decision-making. Some caregivers shared techniques they were able to use to improve communication.

*"I think there should be a study for parents, and they should be educated on how to communicate with their kids. . . they still have that mentality of saying you cannot talk about sex with your child".*

Female, 18 years old, sexually active, and in multi-partner relationships.

*". . . when a child wants to talk to you, they must not be scared to talk to you. . . must have a friendship. . . for example I hugged my child and told her that it is a mistake. . . I am trying to bring her close to me. . . I do not shout . . . The child must not be afraid to talk to you as the parent".*

Female caregiver, 49 years old, married.

## Discussion

Making choices to improve one's sexual health outcomes is the foundation for positive sexual decision-making. Through our qualitative investigation conducted in South Africa, we found that adolescents had adequate knowledge of risks associated with certain sexual behaviours but do not translate this knowledge into choices for healthy sexual outcomes. In this resource-limited setting, poor communication between adolescents and caregivers emerged as a dominant barrier. This barrier limited the type of support that caregivers could provide to their adolescents to enable informed decision-making. While caregivers knew of the benefits of communicating and supporting their adolescents, evidence suggests that lack of knowledge and skills, combined with generational and educational gaps between caregivers and adolescents, also contributed to caregivers' sense of disempowerment, lack of self-efficacy, and reluctance to engage in communication in healthy sexual choices. Communication tools and guidance on how adolescents and caregivers could communicate about sexual matters could create enabling environments to improve adolescents' agency to make informed, healthy decisions regarding their sexual behaviours. This study supports previous reports highlighting the importance and strengthening of caregiver engagement with adolescents in healthy sexual decision-making [11–14, 19, 24]. Strategies include providing a supportive attitude, being able to monitor adolescent behaviour, and improving family dynamics where both caregivers serve important roles in preventing adolescents engaging high-risk sexual behaviours [11–14, 19, 24].

Adolescents and caregivers had contrasting views on their ability to communicate about sexual topics. Some reflected on the healthy relationships they had built in supportive roles.

Others believed this was simply not possible or uncommon practice due to the traditional taboo nature in discussing sexual topics with children. Adolescents expressed limited caregiver trust, caregiver judgemental attitudes and being misunderstood as common barriers in engaging with caregivers. Caregivers knew the benefits of being more supportive but felt helpless as they lacked the knowledge and ability to engage in sexual behaviour discussions with their adolescents. Similar findings have been reported in SSA, where the ability of caregivers to engage with adolescents in sexual health topics are strained due to caregivers' inabilities to discuss sex, generation gaps, proscriptive socio-cultural beliefs and moralistic and religious views [12–14, 25]. Targeted interventions engaging with caregivers to strengthen skills and comfort in having sex topic discussions and providing support for their children in their sexual health choices is a pressing need in SSA and South Africa [12–14, 25]. A South African study identified that increased positive caregiving and caregiving supervision were protective factors to adolescent engaging in HIV risk behaviours [26]. Further, caregiver support could enhance adolescent self-efficacy and improve self-esteem toward healthy sexual decision-making [19].

The lack of support from male caregivers within the family was another factor contributing to a breakdown in communication. The role male caregivers provide is increasingly being challenged, shifting from the stereotypical breadwinner role, to being supportive and understanding toward their adolescent children [13, 14, 27]. Other male caregivers, including uncles, brothers, grandfathers, or other adult males also have an important supportive role to serve [27]. In our study, male caregivers desired to provide more support to their adolescents yet seemed insecure in this role. Reframing the stereotypical male caregiver role, towards empowering men to be understanding and supportive toward their children could address this barrier [24, 27]. In this study adolescent boys expressed their inability to reach out to either male or female caregiver, amplifying the need to intervene for adolescent boys to improving healthy decision-making. Interventions focused in targeting male caregivers have been successful in improving overall health and social outcomes at home [28]. Interventions with targeted approaches in improving male caregiver skills to nurture relationships with children, improving nuclear family relationships, empowering males with knowledge and skill to refrain from violence at home, and engaging men as partners in their female partners sexual reproductive and maternal health have produced encouraging results. Increasing male caregiver shared responsibility in domestic chores and caregiving of children have improved caregiver gender-equality inadequacies, enhances nuclear family relationships and have reduced male dominance in decision-making [28].

Peer pressure and desires to conform to activities their peers engage in were other influences in adolescent choices to engage in sexual behaviours. The drive for materialistic rewards, particularly if they are poor, influences adolescent engagement in transactional sexual relationships [29, 30]. Friends, peers and sexual partners can be influential in adolescents engaging in sexual behaviours, sexual abuse, alcohol and illicit drug use. While some participants reflected on positive influences friends, peers and sexual partners can have, dominant concern to the negative influences was shared by both adolescents and caregivers. These findings provide insights to the agency adolescents have in choosing to engage in sexual behaviours. Across SSA, adolescents' risks for becoming sexually active is not by choice alone and have multiple intrinsic and extrinsic influences [31, 32]. While adolescents conform less to traditional and cultural gender roles, relationship power inequities and hegemonic masculine beliefs continue to fuel beliefs in male dominance [32]. All of which have poor outcomes for adolescents, specifically AGYW, in defining their agency toward positive choices for their sexual and reproductive health [32]. The evidence provided here and in literature resonates the need for enhanced community and social support systems that transcend across SEM levels for enduring behaviour change [19].

Another strategy to improve translating knowledge toward healthy sexual decision-making, is in using contextually relevant gain-framed messaging to emphasize the benefits of engaging in behaviours which have positive health outcomes [33]. Although adolescents receive information and have knowledge of sexual behaviours, they struggle to make immediate healthy choices for their sexual health. The standard educational material provided to adolescents possibly does not emphasise the benefits for individual well-being based on immediate choices around their sexual behaviour. Gain-framed messaging (benefits of healthy sexual decision choices) instead of loss-framed messaging (harms of not making healthy sexual decision choices) can have motivating results in healthy sexual decision choices [33, 34]. In addition, reaching adolescents and caregivers through television and social media has broader potential in propagating gain-framed messaging [35]. The role of gain-framed messaging in promoting healthy sexual decision-making is not known in SSA [35] and represents opportunity to investigate this strategy in future.

Our study was limited to participants who were already accessing services in a research setting, and an unequal representation of both male adolescents and caregivers. Due to the design and provisions of the parent study, the data collected for this study was limited to four FGDs, and therefore we cannot confirm nor conclude saturation of emergent themes. We also did not include the perspectives of external stakeholders such as community leaders, policy makers, and potential influential role models within the community. Thus, this study is restricted in its ability to capture the perspectives of the broader community, including capturing the perspectives of more adolescents and caregivers, and the findings should be considered within these limitations. We are not able to determine the degree of community proscription in having sexual behaviour discussions between adolescents and caregivers, and if this is contextually related to specific community groups. Future studies could include the voices of more adolescent and caregiver participants, and the inclusion of other members of communities and societies.

Despite these limitations, our study has strengths in that it highlights how vital caregiver-adolescent communication could be in informing healthy decision-making among adolescents toward healthy sexual and reproductive health outcomes. This study is among a few studies highlighting the perspectives of adolescents and their caregivers that influences adolescent sexual decision-making and provides valuable insights into strategies that could be applied in future adolescent behavioural interventions. While the data was collected in 2018, the continued HIV burden among AGYW represents the importance to understand factors that may continue to influence adolescent sexual decision-making and to inform prevention strategies that can be used to reduce adolescents' exposure for HIV [1–5].

## Conclusions

Overall, the study provides context of the multiple individual and interpersonal factors influencing adolescent sexual decision-making and behaviour, and the need to provide a supportive environment which enables sexual behaviour choices for healthy sexual and reproductive outcomes. Such supportive environments include protective measures which seeks to prevent marginalizing sexual behaviour choices at home and in community, and enables adolescents the agency to make the choices with knowledge of behaviours which are likely to have positive health sexual and reproductive outcomes [36]. The study provides valuable insights to the critical role caregiver-adolescent communication has in promoting healthy sexual decision-making. The findings underscore the importance of implementing targeted, culturally relevant interventions which include sexual education and skill package to equip both male and female caregivers to engage in conversations on sexual behaviour. The study as well highlights the

potential in including gain-framed messaging within intervention packages which could enable adolescents' agency to translate knowledge towards healthy sexual choices. Lastly, the study adds to existing evidence highlighting the potential benefits in engaging men in improving their nurturing and gender-equitable roles at home, and the powerful response they can have in improving health and social outcomes for their family.

## Supporting information

**S1 Text. Adolescent FGD guide.**
(PDF)

**S2 Text. Caregiver FGD guide.**
(PDF)

**S1 Checklist. COREQ checklist.**
(PDF)

## Acknowledgments

We thank the study participants, and the staff at The Aurum Institute Rustenburg Clinical Research Site for expert implementation of the main study. Specifically, Ireen Mosweu for coordinating the recruitment and retention activities, and Octavia Madikwe for her clinical insights.

## Author Contributions

**Conceptualization:** Heeran Makkan, Candice Chetty-Makkan.

**Data curation:** Funeka Mthembu, Omphile Masibi, Thuso Molefe, Candice Chetty-Makkan.

**Formal analysis:** Heeran Makkan, Yvonne Wangui Machira, Candice Chetty-Makkan.

**Funding acquisition:** Pholo Maenetje.

**Investigation:** Heeran Makkan, Yvonne Wangui Machira, Candice Chetty-Makkan.

**Methodology:** Heeran Makkan, Pholo Maenetje, Matt A. Price, Candice Chetty-Makkan.

**Resources:** Funeka Mthembu, Pholo Maenetje.

**Supervision:** Funeka Mthembu, Vinodh Aroon Edward.

**Writing – original draft:** Heeran Makkan, Yvonne Wangui Machira, Matt A. Price, Vinodh Aroon Edward, Candice Chetty-Makkan.

**Writing – review & editing:** Yvonne Wangui Machira, Funeka Mthembu, Omphile Masibi, Thuso Molefe, Pholo Maenetje, Vincent Muturi-Kioi, Matt A. Price, Vinodh Aroon Edward, Candice Chetty-Makkan.

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
