## [Decision Letter · Decision Letter 0]

12 Aug 2024

PGPH-D-24-01646

Adolescent choices and caregiver roles: Understanding individual and interpersonal influences on sexual decision-making in South Africa

Dear Dr. Makkan,

Thank you for submitting your manuscript to PLOS Global Public Health. After careful consideration, we feel that it has merit but does not fully meet PLOS Global Public Health’s publication criteria as it currently stands. Therefore, we invite you to submit a revised version of the manuscript that addresses the points raised during the review process.

Editor comments:

The reviewers and I found the manuscript to be generally well-written, and feel it provides interesting insights into the dynamics of caregiver/youth conversations and education concerning sexual health.However, there are a few areas that could use some additional clarification and expansion. In particular, additional information on the qualitative sample (how recruitment occurred, whether the caregivers are linked to the youth participants or not, etc.). In addition, please see and address Reviewer 3's comments concerning the language and areas that should be elaborated on, if possible.

We look forward to receiving your revised manuscript.

Kind regards,

Marie A. Brault, PhD

Academic Editor

Journal Requirements:

Additional Editor Comments (if provided):

Reviewers' comments:

Reviewer's Responses to Questions

**Comments to the Author**

1. Does this manuscript meet PLOS Global Public Health’s publication criteria? Is the manuscript technically sound, and do the data support the conclusions? The manuscript must describe methodologically and ethically rigorous research with conclusions that are appropriately drawn based on the data presented.

Reviewer #1: Partly

Reviewer #2: Yes

Reviewer #3: Yes

2. Has the statistical analysis been performed appropriately and rigorously?

Reviewer #1: N/A

Reviewer #2: Yes

Reviewer #3: N/A

3. Have the authors made all data underlying the findings in their manuscript fully available (please refer to the Data Availability Statement at the start of the manuscript PDF file)?

Reviewer #1: No

Reviewer #2: Yes

Reviewer #3: Yes

4. Is the manuscript presented in an intelligible fashion and written in standard English?

Reviewer #1: No

Reviewer #2: Yes

Reviewer #3: Yes

5. Review Comments to the Author

Reviewer #1: Abstract

1. I would advise that you structure the abstract in to four sections i.e. Introduction, Methods, Results and Conclusions

2. Sentence on line 42 and 43 is not clear may be it could be,

3. Thematic analysis revealed that while adolescents had access to sexual education from various sources, this knowledge did not translate into healthy sexual decision-making.

4. The term sexual activity is not clearly understood in the entire manuscript. Does sexual activity mean sexual matters (issues) or intercourse please be clear

5. You should have like a maximum of 5 key words just after the conclusion on the abstract

6. You should move authors’ contributions after the abbreviations’ section

Introduction

1. On line 86. The HIV burden among sub-Saharan African (SSA) adolescents . . . make sub upper case

2. On line 108. The role of discussing sensitive matters . . . what do you mean by sensitive matters? These matters could be family, political, religious, economic, etc. please be clear

3. On line 127. . . a qualitative exploratory study to understand the factors that influences sexual .. . The sentence should read . . . a qualitative exploratory study to understand the factors that influence sexual .. .

4. Please include a geographical scope in your concluding statement of the introduction. E.g. Therefore, we conducted a qualitative exploratory study to understand the factors that influences sexual decision making from the perspectives of adolescents and their caregivers in . . . .

Methods

1. Study design and setting should be study design and settings

2. On lines 137 and 138 you state that, Adolescents between the ages of 12-19 years constitute 14.6% of the population and have a high prevalence of HIV. Please mention this high prevalence.

3. Paragraphs two and three in the methods section are disjointed. They do not speak to the introduction nor to the study title. Please rewrite this section basing on the current objectives not anything else.

4. How many FGDs did you conduct with Adolescents and with Caregivers also describe their composition. From the text one guesses that they were four 2 with adolescents and 2 with caregivers. FGDs being only 2 for each of the categories seem to have compromised validity and reliability of your findings. This being a qualitative study where data collection is collected up to saturation point mere two FGDs with each of the categories makes the findings less dependable.

5. In lines 159 – 161, you state that FGD guides (attached S1 and S2) were used to explore the adolescent and caregiver knowledge of sexual and reproductive health, and factors contributing to sexual and reproductive health behaviour. Yet the objective of this study was to understand the factors that influences sexual decision making from the perspectives of adolescents and their caregivers (refer to lines 126 - 128). This means that you collected somewhat different data from your objective. Please rework on this section very well. Be consistent and stick to the objective of this Manuscript.

6. Please describe the inclusion and exclusion criteria of the adolescents and the care givers in to the study

7. In line 194 – 195 you state that, All participants were reimbursed ZAR150.00 (about $8.20 USD) for participation in the FGDs. This imply means you paid them for taking part in the study. I suggest you revise this statement to mean that all participants were given a small token of appreciation ZAR150.00 (about $8.20 USD) for sparing their time to take part in the study.

Results

1. In line 203 you state 17 adolescents participants in the study but in lines 152 – 154 you state that, Two in-person FGDs were conducted with adolescents and two with caregivers following enrolment into the main study and before a month-3 follow-up visit. Each FGD comprised 8-10 participants. This is a glaring contradiction and poses a number of questions on researcher team’s honesty.

2. Is the data of 2018 still applicable and publishable in the last half of 2024 in order to inform policy or any other development agenda? Haven’t the findings of this study been over taken by events considering the time differences?

Discussion

1. The discussion sections flows very well. However I recommend that the research teams adopts the art of using short sentences as compared to long ones. This makes the reader follow the story clearly without having to keep rereading the sentences.

2. Please proof read the entire manuscript to avoid obvious errors and punctuation issues

3. I also suggest that a section of the study limitations be included in the Manuscript

4. A section on data availability should also be included in the Manuscript

Conclusion

1. This is well articulated and emanates from the study findings.

References

1. These should be the last section of the manuscript there after any appendices can follow

Reviewer #2: This is an important, thorough, and well-written paper. I recommend the following minor revisions:

1. In the abstract section, clarify the statement: “Thematic analysis revealed that while adolescents had access to sexual education from various sources, where the knowledge does not translate into healthy sexual decision-making.”

2. In the study design and setting section, provide more details on the specific methods used for convenience sampling to enhance transparency and replicability.

3. While the discussion section offers valuable insights, it needs improvements for better clarity and depth. Integrate the findings more effectively with existing literature to contextualize the results, such as referencing previous studies that identify similar barriers like poor communication between adolescents and caregivers.

4. The discussion on the intention-behavior gap and gain-framed messages requires more detailed explanations of practical implementation strategies. Additionally, the section on male caregivers should include specific interventions or programs that have successfully redefined traditional roles and how these could be adapted in the study context.

5. The limitations section should be more thorough, discussing the potential impacts of these limitations on the findings and suggesting ways to address them in future research. Propose strategies for including a more diverse participant pool or involving external stakeholders to provide a more comprehensive understanding.

6. The conclusion effectively summarizes the key findings of the study but could be enhanced for clarity and impact. Here are some suggestions:

o Emphasize practical implications by highlighting how the findings can be directly applied to develop specific interventions or programs.

o Include future research directions to suggest areas for future research to build on the study's findings.

Reviewer #3: Review of manuscript:

Adolescent choices and caregiver roles: Understanding individual and interpersonal influences on sexual decision-making in South Africa

This is a well written paper overall, and addresses an important topic. I have a few specific comments and some more general ones for the authors’ consideration. I have also included some suggestions for additional literature which I think could help to strengthen some of the areas of the discussion.

Specific comments:

Line 35: “South African adolescents are at-risk for HIV infection” – through sexual transmission. Consider adding this clarification, otherwise the next sentence feels slightly non secateur

Line 37: The adolescents’ caregivers? Were they paired?

Line 42: something missing in this sentence? – is incomplete: “where the knowledge does not translate into healthy sexual decision-making….??”

Line 364: what is a “good choice” ? perhaps a risk of being morally laden by suggesting that choices are good or bad. Perhaps rephrase to focus on the outcomes of the choice resulting in positive or negative health outcomes.

Line 366: “knowledge of sexual behaviour” ? or knowledge of risks associated with sexual behaviours and risk reduction / avoidance / prevention strategies?

Line 369: enabled informed decision making?

Lines 370-371: more than just their ability to communicate effectively about SRH - evidence suggests that lack of knowledge and skills, combined with generational and educational gaps between parents and adolescents, also contribute to parents’ sense of disempowerment, lack of self- efficacy, and reluctance to discuss SRH.

Line 388: what is meant by “caregiver intervention” ?

General comments:

In the findings section, the authors outline factors at the individual and interpersonal / relationship levels of the SEM framework. It would be interesting to know whether factors at the socio-cultural / contextual level came up – as social and cultural guidelines and proscriptions around sexuality communication between adolescents and caregivers are very deeply rooted, particularly in the sub-Saharan African setting.

Peer pressure – the authors describe the influence of social media etc – but it would seem that some of this relates to peer pressure, desire to conform, and perceptions of what other adolescents are doing.

Related to this - what about adolescents’ desire to be fashionable etc. Evidence suggests that materialism and pressure to conform and achieve social status influences adolescent risk behaviours.

One aspect that could be expanded on pertains to the ways in which power inequities and agency disparities impact the ability of adolescent girls to make decisions about their SRH and t op adopt behaviours that are protective for their sexual health.

Relating to my comment about what constitutes a “good choice” – in my opinion, the authors could pay closer attention to the language and terminology used around risk – to ensure that the language avoids any moralising. There is literature about the ‘discourse of risk’ (see the Shoveller reference below). I would urge the authors to consider the framing of adolescent choices, risk engagement and decision making to ensure that the focus is on health outcomes.

Suggested literature:

Duby, Z., Jonas, K., McClinton Appollis, T. et al. From Survival to Glamour: Motivations for Engaging in Transactional Sex and Relationships Among Adolescent Girls and Young Women in South Africa. AIDS Behav (2021). https://doi.org/10.1007/s10461-021-03291-z

Leclerc Madlala S. Transactional sex and the pursuit of modernity. Soc Dyn. 2003;29(2):213–33. https://doi.org/10.1080/02533 950308628681.

Zembe YZ, Townsend L, Thorson A, Ekström AM. “Money talks, bullshit walks” interrogating notions of consumption and survival sex among young women engaging in transactional sex in post-apartheid South Africa: a qualitative enquiry. Glob Health. 2013;9(28):1–16. https://doi.org/10.1186/1744-8603-9-28.

Jean A. Shoveller Assistant Professor PhD & Joy L. Johnson (2006) Risky groups, risky behaviour, and risky persons: Dominating discourses on youth sexual health, Critical Public Health, 16:1, 47-60, DOI: 10.1080/09581590600680621

Duby, Z., Bergh, K., Jonas, K. et al. “Men Rule… this is the Normal Thing. We Normalise it and it’s Wrong”: Gendered Power in Decision-Making Around Sex and Condom Use in Heterosexual Relationships Amongst Adolescents and Young People in South Africa. AIDS Behav (2022). https://doi.org/10.1007/s10461-022-03935-8

6. PLOS authors have the option to publish the peer review history of their article (what does this mean?). If published, this will include your full peer review and any attached files.

**Do you want your identity to be public for this peer review?** For information about this choice, including consent withdrawal, please see our Privacy Policy.

Reviewer #1: **Yes: **Ronald Bahati

Reviewer #2: No

Reviewer #3: No

---

## [Decision Letter · Decision Letter 1]

13 Nov 2024

Adolescent choices and caregiver roles: Understanding individual and interpersonal influences on sexual decision-making in South Africa

PGPH-D-24-01646R1

Dear Mr Makkan,

We are pleased to inform you that your manuscript 'Adolescent choices and caregiver roles: Understanding individual and interpersonal influences on sexual decision-making in South Africa' has been provisionally accepted for publication in PLOS Global Public Health.

Best regards,

Marie A. Brault, PhD

Academic Editor

Reviewer Comments (if any, and for reference):

Reviewer's Responses to Questions

**Comments to the Author**

1. If the authors have adequately addressed your comments raised in a previous round of review and you feel that this manuscript is now acceptable for publication, you may indicate that here to bypass the “Comments to the Author” section, enter your conflict of interest statement in the “Confidential to Editor” section, and submit your "Accept" recommendation.

Reviewer #1: All comments have been addressed

Reviewer #2: All comments have been addressed

2. Does this manuscript meet PLOS Global Public Health’s publication criteria? Is the manuscript technically sound, and do the data support the conclusions? The manuscript must describe methodologically and ethically rigorous research with conclusions that are appropriately drawn based on the data presented.

Reviewer #1: Yes

Reviewer #2: Yes

3. Has the statistical analysis been performed appropriately and rigorously?

Reviewer #1: N/A

Reviewer #2: Yes

4. Have the authors made all data underlying the findings in their manuscript fully available (please refer to the Data Availability Statement at the start of the manuscript PDF file)?

Reviewer #1: Yes

Reviewer #2: Yes

5. Is the manuscript presented in an intelligible fashion and written in standard English?

Reviewer #1: Yes

Reviewer #2: Yes

6. Review Comments to the Author

Reviewer #1: The Authors responded to all comments raised in my first review and i hereby recommend the paper to be accpeted for publication

Reviewer #2: (No Response)

7. PLOS authors have the option to publish the peer review history of their article (what does this mean?). If published, this will include your full peer review and any attached files.

**Do you want your identity to be public for this peer review?** For information about this choice, including consent withdrawal, please see our Privacy Policy.

Reviewer #1: No

Reviewer #2: No
